# Hematological and coagulation alterations and splenic response following *Macrovipera lebetina obtusa* envenomation: Evaluation of ovine-derived experimental antivenom

Gevorg Avagyan[1,2]*, Vardan Dabaghyan[3], Heghine Khachatryan[2], Ashot Aslanyan[4], Arsen Kishmiryan[4], Anna Karapetyan[1], Naira Ayvazyan[4]

1 Department of Human and Animal Physiology, Yerevan State University, Yerevan, Armenia, 2 Scientific Division, Yeolyan Hematology and Oncology Center, Yerevan, Armenia, 3 Department of Pathological Anatomy, Yerevan State Medical University after Mkhitar Heratsi, Yerevan, Armenia 4 Department of Toxicology, Orbeli Institute of Physiology, National Academy of Sciences of the Republic of Armenia, Yerevan, Armenia,

* drgevavagyan@gmail.com

## Abstract

*Macrovipera lebetina obtusa (M. l. obtusa)* is the most medically significant viper species in Armenia and the region, responsible for the majority of snakebite cases. This study investigated the hematological, coagulation, and histopathological effects of *M. l. obtusa* envenomation in rats and mice, and evaluated the therapeutic efficacy of an ovine-derived experimental antivenom. Hematological analysis revealed significant increases in red blood cell (RBC) count and hemoglobin (HGB) concentration following envenomation, suggestive of hemoconcentration likely due to vascular leakage. Coagulation studies demonstrated marked prolongation of prothrombin time (PT) and activated partial thromboplastin time (APTT), indicating venom-induced coagulopathy. Notably, PT was more severely affected, and its elevation persisted even after antivenom administration, suggesting incomplete neutralization of venom activity. Histopathological examination of spleen and skin tissues showed progressive structural disruption, including hemorrhage, edema, and lymphoid follicle hyperplasia, which remained evident up to seven days post-envenomation. While the experimental antivenom provided partial systemic protection and improved some hematological parameters, it was unable to reverse the venom-induced tissue and coagulation abnormalities fully. These findings highlight the complex pathophysiology of *M. l. obtusa* venom and underscore the need for adjunctive therapies targeting vascular integrity and immune regulation in the management of viper envenomation.

**Data availability statement:** All relevant data are in the manuscript and its supporting information files.

**Funding:** This study was supported by the Higher Education and Science Committee of the Ministry of Education, Science, Culture and Sports of RA (#21AG-1F031 to NA). The funders had no role in study design, data collection and analysis, decision to publish, or preparation of the manuscript.

**Competing interests:** The authors have declared that no competing interest exists.

## Author summary

The blunt-nosed viper *M. l. obtusa* (*Macrovipera lebetina obtusa*) is the most dangerous snake in Armenia and surrounding areas, responsible for most local snakebite cases. Its venom can cause serious blood and tissue damage, which may not be fully reversed even with antivenom. In this study, we examined how the venom affects blood cells, blood clotting, and organ structure in laboratory rats and mice. We also tested an experimental antivenom made from sheep antibodies (ovine-derived) to see how well it could protect against these effects. The venom caused dehydration of the blood (hemoconcentration), likely due to leaky blood vessels, and triggered strong clotting disturbances. One clotting measure, the prothrombin time, stayed abnormal even after antivenom treatment, showing that the venom's effects were only partly neutralized. Microscopic tissue analysis revealed bleeding, swelling, and immune-related changes in the spleen and skin that lasted for at least a week. While the ovine-derived antivenom improved some blood test results, it could not fully prevent tissue damage or restore normal clotting. These findings show why managing viper bites can be challenging and suggest that additional treatments may be needed alongside antivenom to protect patients.

## 1. Introduction

Even in the 21st century, snakebite envenoming remains a significant global public health burden [1,2]. According to the World Health Organization (WHO) reports, an estimated 5.5 million snakebites occur worldwide yearly, resulting in more than 2.5 million cases of envenomation. These incidents contribute to approximately 81,500–138,000 deaths annually, with many survivors experiencing long-term complications such as limb amputations and permanent disabilities. The WHO underscores that the majority of these adverse outcomes are preventable through improved availability and accessibility of safe, high-quality antivenoms, which constitute the most effective therapeutic intervention for neutralizing the toxic effects of snake venom [3,4]. Conceptually unchanged for over 130 years, these therapies are associated with various issues. Some significant limitations of the existing antivenoms are low effectiveness due to a large proportion (~80–90%) of immunoglobulins not targeting venom toxins; frequent adverse reactions from administering large doses of foreign immunoglobulins; the necessity of intravenous administration in a healthcare setting, which is not always able to neutralize local tissue damage in cases of viper bites [5,6]. Although the effects of different venoms and purified snake venom toxins on the prey organism are well documented in many articles, investigations into the organism's reaction to antivenom injection and the cause of the high percentage of side effects for this treatment are scarce [7,8].

Snake venoms are a cocktail of proteins and peptides, which make up approximately 90–95% of the venom's dry weight, along with minor components. Four

abundant toxin families are prevalent across various snake venoms and serve as key targets for toxin-inhibiting therapeutics: phospholipases A2 (PLA2s), snake venom metalloproteinases (SVMPs), snake venom serine proteases (SVSPs), and three-finger toxins (3FTxs) [9]. The primary clinical manifestations of snakebite envenomation are typically classified into three major syndromes: haemotoxic effects (such as hemorrhage and coagulopathy), neurotoxic effects (including muscle paralysis), and cytotoxic effects (such as localized tissue necrosis) [10,11]. Haemotoxic effects are a predominant feature of envenomation, particularly after bites from viperid snakes (family Viperidae), and are primarily induced by snake venom metalloproteinases (SVMPs), snake venom serine proteinases (SVSPs), and phospholipases A2 (PLA2s). [12,13]. SVMPs are well known for their hemorrhagic effects and their capacity to disrupt multiple stages of the blood coagulation cascade, leading to a dangerous combination of systemic bleeding and impaired blood clotting in prey and victims. [14,15]. Research has shown that SVMP-induced hemorrhage occurs through a two-step mechanism [16,17]. Initially, SVMPs degrade the basement membrane and adhesion proteins within the endothelial cell-matrix complex, disrupting capillary stability. In the subsequent phase, endothelial cells detach from the basement membrane and become markedly thinner, leading to capillary wall destruction and blood extravasation. Beyond their proteolytic function, snake venom metalloproteinases (SVMPs) contribute to hemostatic alteration by interfering with the coagulation cascade, thereby exacerbating hemorrhagic outcomes [18]. While SVMPs are predominantly associated with capillary vessel degradation, snake venom serine proteinases (SVSPs) primarily exert their pathological effects through targeted disruption in blood coagulation (either procoagulant or anticoagulant), fibrinolysis, platelet aggregation, and blood pressure regulation, potentially resulting in life-threatening complications for snakebite victims [19,20]. Venoms can influence blood coagulation by either accelerating the process (procoagulation) or impairing it (anticoagulation) [21]. Coagulopathy is a major contributor to both morbidity and mortality in snakebite patients, either as a direct consequence of envenomation or through secondary complications. A wide variety of venom components can act as procoagulants, causing *in vivo* activation of the coagulation system. However, in most cases, this does not result in massive thrombosis and consequent embolic disease, but rather causes consumption of coagulation factors, resulting in clinical anticoagulation [22].

In Armenia, 22 snake species are known to inhabit the region, four of which are venomous vipers. Two of these—the Armenian steppe viper (*Vipera eriwanensis*) and the Darevskii viper (*Vipera darevskii*)—are both rare and small, delivering limited venom and posing little danger to people. The Armenian viper (*Montivipera raddei*), although venomous, is rarely encountered due to its limited habitat and protected status. The most dangerous species in Armenia is the blunt-nosed viper *Macrovipera lebetina obtusa* (*M.l. obtusa*), which is widespread throughout the country, including in cities and the capital [23,24]. Due to its extensive range, this snake is also a significant cause of snakebites across the South Middle East and the Caucasus region.

In the current investigation, we aim to explore how *M. l. obtusa* snake venom affects blood function and spleen activity to uncover new details about the biological processes triggered by envenomation. By comparing systemic blood responses with splenic function, we are trying to identify specific hematological disruptions and physiological adaptations following envenomation. The research also deciphered how these responses evolve during the antivenom treatment, demonstrating a clearer view of how the body reacts and recovers. Therefore, the results could help develop more effective treatment strategies for snakebites, particularly for those bitten by *M. l. obtusa* snakes.

## 2. Materials and methods

### 2.1. Ethics statement

All experimental procedures were conducted in accordance with the principles of laboratory animal care as approved by the Bioethics Committee of Yerevan State University (YSU). The study complied with the European Convention for the Protection of Vertebrate Animals Used for Experimental and Other Scientific Purposes (Directive 2010/63/EU of the European Parliament and the Council, September 22, 2010).

## 2.2. Equipment and reagents

Auto Hematology Analyzer URIT-3000Vet Plus was used for CBC (complete blood count); for coagulation testing - Wondfo Optical Coagulation Analyzer (OCG-102) along with the Prothrombin time reagent kit (PT and International Normalized Ratio (INR)) and Activated Partial Thromboplastin Time reagent kit (APTT). Chemicals for buffers and solutions, buffered formalin, xylene, ethanol were obtained from Medisar LLC (Armenia). Crude venom of *M. l. obtusa* was sourced from the specimens housed in the serpetarium of the Orbeli Institute of Physiology, National Academy of Sciences (Yerevan, Armenia), vacuum dried at ambient temperature, and stored in the dark at 4°C until use.

## 2.3. Experimental antivenom

The processing and development of ovine-derived experimental monovalent antibody serum against *M. l. obtusa* venom were performed by purifying and concentrating the immunoglobulins using caprylic acid, as described elsewhere [25]. Briefly, the animals were immunized with subcutaneous injections of increasing doses of venom from the *M. l. obtusa* snake in two stages. During the first stage, the animals were injected with venom dissolved in adjuvant (sodium alginate) to reduce the risk of tissue damage. The injections were made in two anatomical regions on the animal's neck, close to the axillary lymph nodes. The second stage of immunization was performed with *M. l. obtusa* venom without adjuvant; a 2-week break was taken to allow the organism to produce sufficient antibodies before blood harvesting. The serum containing IgG antibodies was subsequently processed with caprylic acid to separate the antibodies, followed by further ultrafiltration using the Sartoflow Smart Small-Scale Benchtop TFF System (Sartorius, USA). The titration of the purified antibodies from sheep sera was checked by immunofixation capillary electrophoresis (Helena Biosciences Europe, UK) for each batch of IgGs.

## 2.4. Animal maintenance

Experimental animals (nonlinear gray 12-week-old adult mice weighing 20–25 g and white rats weighing 220–250 g, mixed sex) were acclimatized for a minimum of one week before experimentation with their health monitored daily. All experimental animals were maintained under controlled conditions: a 12-hour light/dark cycle (07:00–19:00), stable temperature, and standard feeding. Animals were euthanized after venom injection in compliance with ethical standards.

## 2.5. Animals survival assays via a preincubation model

The LD50 of *M. l. obtusa* venom ($18.4 \pm 1.4$ µg/mouse, i.v., and $1.86 \pm 0.1$ µg/kg, i.p.) was used according to previous studies [26,27]. The mice were divided into three groups for spleen histological analysis: a) healthy control; b) mice injected intravenously with freshly diluted *M. l. obtusa* venom at a lethal dose of 5 $LD_{50}$ ($5 \times 18.4$ µg); c) mice injected intravenously with a preincubated mixture of venom and antivenom. Each mouse received 0.5 mL of the corresponding injection. Group b (venom only) received 92 µg of *M. l. obtusa* venom diluted in 0.5 mL saline. Group c (venom + antivenom) received a mixture containing 92 µg of *M. l. obtusa* venom in 250 µL of saline combined with 250 µL of antivenom, corresponding to approximately 13 mg of IgG [25]. The venom-antivenom mixture was preincubated for 30 minutes at 37 °C. The rats were also divided into three groups to assess hematological and coagulation alterations: a) healthy control; b) rats injected intraperitoneally with freshly diluted *M. l. obtusa* venom at an $LD_{50}$ dose of $1.86 \pm 0.1$ µg/kg; c) rats injected intraperitoneally with a preincubated mixture of venom and antivenom. During the first 24 hour after injections mice were observed to record the lethal cases, as it is recommended by WHO guidelines for the intravenous test [28,29].

## 2.6. Assay of hemorrhagic activity and its neutralization via a preincubation model and a 'rescue' model of envenoming

The hemorrhagic activity was evaluated according to experimental details described by T. Kurtovic et al [26] with some modifications. To assess the local tissue effects of *M. l. obtusa* venom and the protective action of antivenom, rats were

divided into four groups: a) healthy control b) rats injected intradermally with 50 μg of *M. l. obtusa* venom diluted in 100 μL of physiological saline c) rats injected intradermally with a preincubated mixture of 50 μg of venom and antivenom (1:1 ratio, 50 μL venom + 50 μL antivenom), d) rats first injected intradermally with 50 μg of *M. l. obtusa* venom (in 100 μL saline), followed 10 minutes later by a separate intradermal injection of antivenom at the same site. Injections were administered into the shaved dorsal skin using a 30G × ½ insulin syringe. Animals were monitored to check for symptoms of systemic envenoming, as well as for local reactions such as erythema, edema, hemorrhage, and necrosis for up to 24 hours. Hemorrhagic lesions caused by snake venom are characterized by the surface area of hemorrhage observed on the inner side of the skin 24 hours after injection. After the skin lesions were excised, their inner surface was photographed and measured. The images of all samples are then scaled and combined into a single image, after which the affected areas are outlined and pixelated (Adobe Photoshop 2020 was used for image processing and analysis). Thus, the lesion area is represented by pixels, and the average pixel count for each dose is used in the calculations. Results are expressed as NHA (%) = [(PV – PV AnV)/ PV] x 100 ± SE, where NHA is a Neutralization of Hemorrhagic Activity (%), PV represents the mean value of lesion surfaces induced by the venom alone, and PV AnV the mean value of lesion surfaces induced by the venom pre-mixed with antivenom or after the local antivenom administration, followed by the venom injection.

## 2.7. Tissue processing and Hematoxylin and Eosin (H&E) staining

In the process of dissection and tissue processing, the classical technique to prepare morphological slides was employed. Samples extracted from spleen and skin underwent careful dissection before being delicately embedded in paraplast. Utilizing a rotary microtome, the paraffin blocks were precisely sectioned into slices measuring 2.5 μm in thickness. These meticulously crafted sections were then meticulously mounted into glass slides, ready for detailed histological examination. This method, rooted in classical histological practices, ensures the preservation of tissue integrity and allows for accurate analysis of cellular structures and pathological changes [30].

The histopathological examination of envenomed animals' spleen and skin were carried out following the classical method [30]. At the conclusion of the *in vivo* tests (i.e., 24 h, 3 & 7 days), the surviving mice were euthanized, and their organs were removed and preserved in 10% buffered formalin. All tissues were dehydrated in a graded series of ethanol through 70, 80, 90, 95, and 100% for 1 h each. After two changes of xylene dehydration for 1 h each specimen was processed for embedding in paraffin. The embedded samples were cut into cross-sections and stained with H&E. The tissues were examined and photographed under magnifications ×100 and ×400 using a light microscope Olympus BX43 (Oympus, Japan).

## 2.8. Hematological procedure

In all rats, blood was collected via cardiac puncture and transferred into hematological vacutainers containing EDTA as an anticoagulant for complete blood count (CBC) analysis, and into tubes containing sodium citrate for coagulation testing [31]. For coagulation samples requiring a total volume of 0.5 mL, the citrate-to-blood ratio was maintained at 1:9 by adding 50 μL of 3.2% sodium citrate and 450 μL of whole blood [31]. All tubes were gently inverted several times to ensure proper mixing and prevent clotting. To estimate the hematological parameters of peripheral blood, an Urit-3000Plus veterinary hematological analyzer was used. Reference Intervals of Hematological Parameters in Rats were used in accordance of manufacturer calibration and in agreement with specialized literature [32,33]. For coagulation testing, a Wondfo Optical Coagulation Analyzer (OCG-102) was used to measure prothrombin time (PT-INR) and activated partial thromboplastin time (APTT) using 20 μL of whole blood per test.

## 2.9. Statistical analysis

All quantitative data are presented as mean ± standard error of the mean (SEM). Differences between groups were evaluated using one-way analysis of variance (ANOVA) followed by Dunnett's post hoc multiple comparison test, comparing each experimental group with the corresponding control group. A two-tailed p-value < 0.05 was considered statistically

significant. Data analysis and graph generation were performed using GraphPad Prism version 8.0.1 for Windows (Graph-Pad Software, La Jolla, CA, USA).

## 3. Results

### 3.1. Changes in Complete Blood Count following *Macrovipera lebetina obtusa* venom exposure and antivenom treatment

The venom of the *M. l. obtusa* snake exerted a significant and consistent impact on the hematological profile of rats, specifically targeting the cellular components of blood as reflected in the Complete Blood Count (CBC). This study meticulously examined alterations in hemoglobin (HGB) and red blood cell (RBC) counts following venom exposure and subsequent administration of antivenom (AnV). The core finding regarding the CBC parameters is a clear and consistent induction of hemoconcentration in both the venom-treated and antivenom-treated groups (Table 1). Across all animals (n = 6) exposed to *M. l. obtusa* venom, a notable and statistically significant elevation in both HGB and RBC levels is observed when compared to the control group, which maintains stable and normal hematological values. This increase strongly suggests a disturbance in fluid balance within the circulatory system, potentially driven by venom-induced vascular permeability leading to plasma leakage, and a consequent relative increase in the concentration of red blood cells and hemoglobin.

Alternatively, the venom may trigger a physiological response such as splenic contraction, causing a release of stored erythrocytes into circulation, thereby elevating RBC count and HGB concentration. It is important to note that the observed timeframe of the study likely precludes a significant contribution from erythropoiesis—the process of new red blood cell production—as this is a more prolonged physiological mechanism.

Similar to the venom group, the antivenom-treated animals also show increased HGB and RBC levels. Although these values remain higher than those in the control group, HGB is consistently lower than in the venom-only group. This pattern suggests that the antivenom exerts a partial neutralizing effect on the venom's impact on fluid balance or splenic function. Nonetheless, the persistence of elevated HGB and RBC levels implies that neutralization is incomplete—potentially because the antivenom is often unable to fully target each venom component. In direct comparison, the control group establishes a baseline of normal hematological parameters. The venom group starkly contrasts with this baseline, exhibiting a pronounced and consistent increase in HGB and RBC, signifying a clear pathological response to the venom. The antivenom group represents an intermediate severity, demonstrating a therapeutic effect by reducing venom-induced hemoconcentration, yet not fully restoring the hematological profile to that of unexposed control animals (Fig 1).

The administration of *M. l. obtusa* venom and antivenom in rats led to an increase in platelet and granulocyte counts, alongside a decrease in total white blood cells. All hematological values remained within the physiological reference range for rats, indicating that the response was subtle and not clinically alarming. The observed granulocytosis may reflect activation of the innate immune system in reaction to the antivenom. At the same time, the rise in platelet count could be attributed to cytokine-mediated stimulation of thrombopoiesis. The slight decrease in total WBC may result from lymphocyte redistribution or stress-induced suppression, both of which are common during immune modulation. Overall, the findings suggest that while antivenom effectively neutralizes venom components, it may also induce a mild and controlled immune response that subtly alters hematological parameters without surpassing normal physiological limits.

### 3.2. Impact of *M. l. obtusa* venom and antivenom on coagulation parameters in rats

This study explored how *M. l. obtusa* venom, as well as the antivenom administered afterward, influences blood coagulation in rats. Key coagulation markers—Prothrombin Time (PT), International Normalized Ratio (INR), and Activated Partial Thromboplastin Time (APTT) were used to monitor changes in the clotting profile. Rats injected with the venom showed an apparent prolongation of Prothrombin Time when compared to untreated controls. This points to

**Table 1. Complete Blood Count (CBC) Parameters Across Experimental Groups (experimental repeats–6, mean ± SEM; one-way ANOVA with Dunnett's multiple comparison test).**

| Group (*) | Control | Venom | AnV + V |
|---|---|---|---|
| WBC 10*9/L (2,9–15,3) | 4.95 ± 0.24 | 3.16 ± 0.21 | 3.21 ± 0.25 |
| LYM % (63,7–90,1) | 59.7 ± 3.49 | 49.4 ± 6.03 | 47.4 ± 6.9 |
| MID % (1,5–4,5) | 5.26 ± 1.24 | 7.23 ± 1.12 | 6.88 ± 1.05 |
| GRAN% (7,3–30,1) | 35 ± 3.29 | 43.3 ± 5.22 | 45.6 ± 5.95 |
| LYM 10*9/L (2,6–13,5) | 2.17 ± 0.19 | 1.32 ± 0.23 | 1.28 ± 0.37 |
| MID 10*9/L (0,0–0,5) | 0.41 ± 0.19 | 0.16 ± 0.03 | 0.15 ± 0.06 |
| GRAN 10*9/L (0,4–3,2) | 1.65 ± 0.17 | 1.35 ± 0.24 | 1.41 ± 0.31 |
| RBC 10*12/L (5,6–7,8) | 7.4 ± 0.44 | 9.46 ± 0.19 | 9.4 ± 0.18 |
| HGB g/dL (12,0–15,0) | 14.93 ± 0.85 | 20.18 ± 0.48 | 19.52 ± 0.49 |
| HCT % (36–46) | 41.3 ± 2.2 | 53.2 ± 1.62 | 52.2 ± 1.24 |
| MCV fL (53–68,8) | 56.01 ± 0.83 | 57.3 ± 0.36 | 54.78 ± 0.79 |
| MCH pg (16–23,1) | 20.1 ± 0.35 | 21.2 ± 0.19 | 20.4 ± 0.41 |
| MCHC g/dL (30–34,1) | 36 ± 0.38 | 37.1 ± 0.41 | 37.3 ± 0.35 |
| RDW_CV % (11–15,5) | 10.75 ± 0.2 | 11.75 ± 0.47 | 10.75 ± 0.22 |
| RDW_SD fL | 28.76 ± 0.47 | 30.95 ± 1.07 | 28.41 ± 0.55 |
| PLT 10*9/L (450–900) | 581.6 ± 119.7 | 525.5 ± 69.35 | 839 ± 74.81 |
| MPV fL (5,8–6,9) | 5.51 ± 0.22 | 5.4 ± 0.04 | 5.41 ± 0.13 |
| PDW fL | 6.65 ± 0.29 | 6.7 ± 0.06 | 6.8 ± 0.2 |
| PCT % | 0.28 ± 0.07 | 0.27 ± 0.04 | 0.43 ± 0.05 |
| P_LCR % | 3.8 ± 0.84 | 3.5 ± 0.18 | 3.4 ± 0.49 |
| P_LCC 10*9/L | 15 ± 3.62 | 17.8 ± 3.29 | 27 ± 5.09 |

**Abbreviations: WBC** - White Blood Cells, **LYM** - Lymphocytes, **MID** - Mid-sized cells (Monocytes, Eosinophils, Basophils), **GRAN** - Granulocytes, **RBC** - Red Blood Cells, **HGB** - Hemoglobin, **HCT** - Hematocrit, **MCV** - Mean Corpuscular Volume, **MCH** - Mean Corpuscular Hemoglobin, **MCHC** - Mean Corpuscular Hemoglobin Concentration, **RDW_CV** - Red Cell Distribution Width – Coefficient of Variation, **RDW_SD** - Red Cell Distribution Width – Standard Deviation, **PLT** - Platelet, **MPV** - Mean Platelet Volume, **PDW** - Platelet Distribution Width, **PCT** - Plateletcrit, **P_LCR** - Platelet-Large Cell Ratio, **P_LCC** - Percentage of Large Cell Count.

\* Between parentheses presented the respective Reference Intervals of Hematological Parameters in Rats [33,34].

a significant disruption of the extrinsic coagulation pathway. Interestingly, the PT values among venom-exposed rats varied widely; some showed extreme elevations, while others had more moderate changes. This variability may stem from individual differences in how the venom components were metabolized or distributed, or in the sensitivity of each animal to the toxic effects.

An unexpected trend emerged in animals that received both the venom and the antivenom. Although the antivenom is intended to neutralize toxic components, these rats exhibited an even more consistent and pronounced prolongation in Prothrombin Time than those given venom alone. Unlike the venom-only group, where the degree of PT disruption was uneven, the antivenom-treated group showed elevated PT across all animals with less variation. This suggests that while the antivenom may effectively counteract life-threatening systemic toxicity, it does not necessarily correct the clotting dysfunction right away. In fact, it might reveal the full extent of the coagulation disturbance initiated by the venom. The APTT values were also prolonged in both venom and antivenom groups compared to controls. However, APTT prolongation was relatively modest and less variable, suggesting that venom components predominantly target the extrinsic pathway or downstream common pathway rather than causing a profound intrinsic pathway defect (Fig 2).

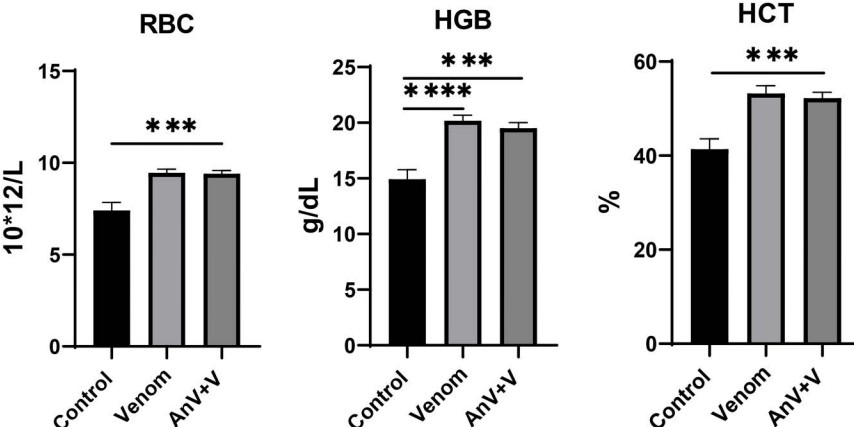

**Fig 1. RBC, HGB, HCT levels after the intravenous administration of PBS (Control), Venom and Antivenom+Venom (AnV+V) (experimental repeats–6, mean±SEM; one-way ANOVA with Dunnett's multiple comparison test, p\*<0.05; p\*\*\*<0.001; p\*\*\*\*<0.0001).**

### 3.3. *M. l. obtusa* envenomation causes certain histopathological changes in spleen tissue even after recovery by antivenom treatment

The spleen tissue sections from the control group mice show a normal histological appearance. The histological structure of the splenic tissue is preserved. The capsule is entirely intact, represented by a thin layer of connective tissue. The lymphoid follicles are developed, and the red pulp is moderately congested. No hemorrhages are observed in the perifollicular areas. The stromal vessels are partially blood-filled (Fig 3A and 3B).

In the venom group, the splenic tissue is edematous. The histological structure is partially preserved. The lymphoid follicles are sufficiently developed. The red pulp is moderately congested. The perifollicular areas contain moderately marked foci of hemorrhage (moderate hemorrhages++). The stromal vessels are congested. The capsule is preserved and is represented by a thin connective tissue (Fig 3C and 3D and Table 2).

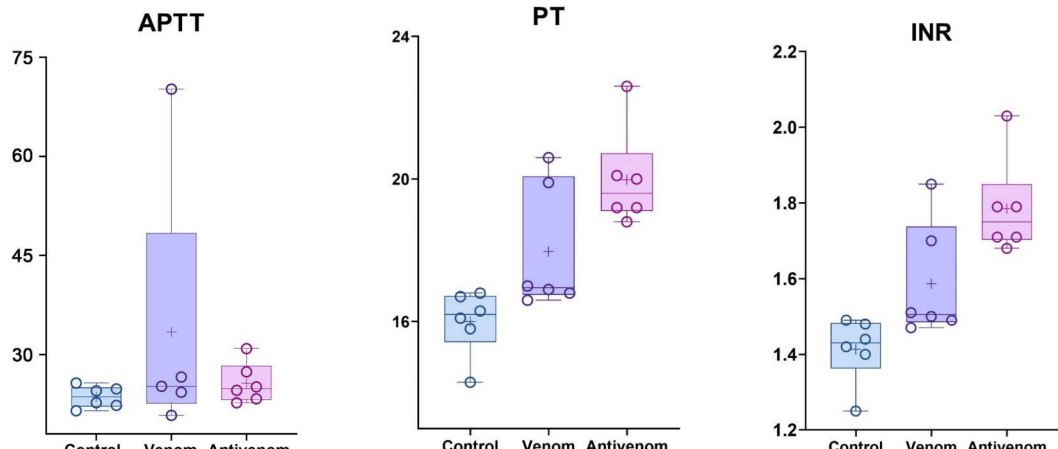

**Fig 2. APTT, PT and INR levels after the intravenous administration of PBS (Control), Venom and Antivenom+Venom (AnV+V) (experimental repeats–6, mean±SEM; one-way ANOVA with Dunnett's multiple comparison test).**

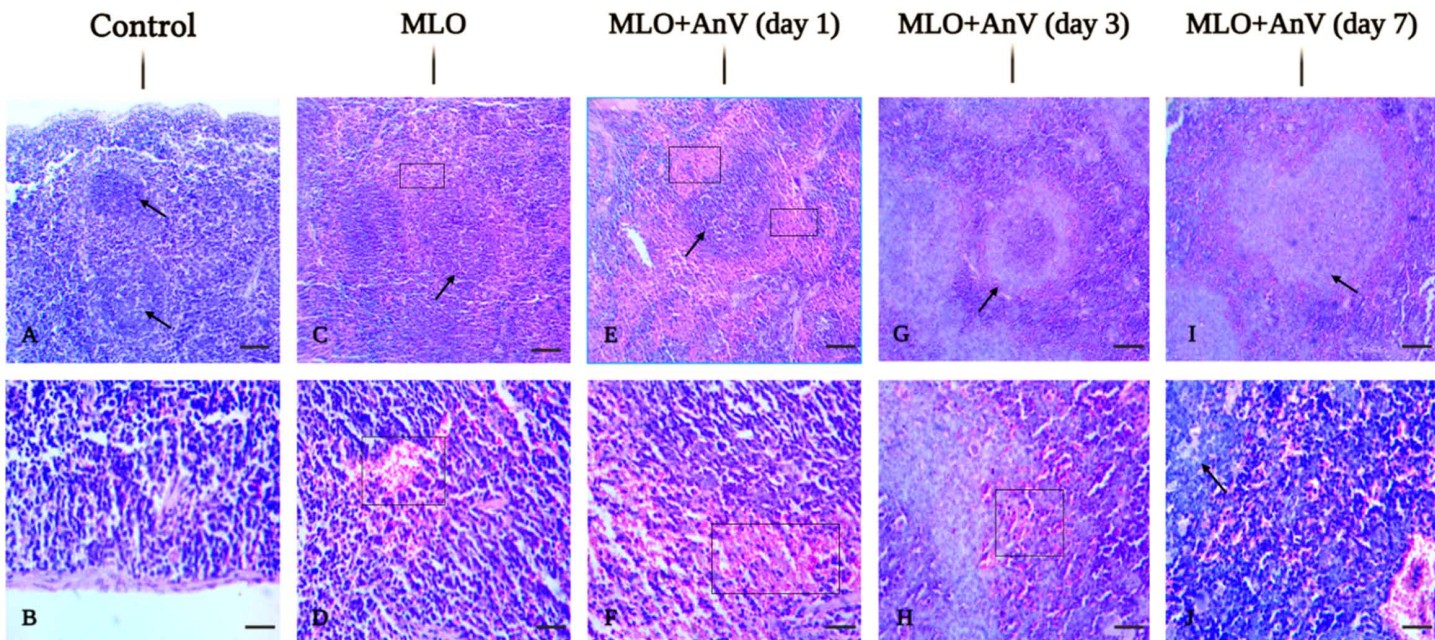

**Fig 3. Histopathological changes in the spleen after injection of *M. l. obtusa* (MLO) venom and venom/antivenom mixture.** The representative field images were acquired with Olympus BX43. Upper row: magnification ×100, scale bar = 250 µm. Lower row: magnification ×400, scale bar = 100 µm. Arrow indicates lymphoid follicle hyperplasia; Square marks areas of hemorrhage.

**Table 2. Tissue damage in Spleen and Skin: + (mild), ++ (moderate), +++ (severe).**

| Group | Spleen | | Skin | |
|---|---|---|---|---|
| | *Lymphoid Follicle Hyperplasia* | *Hemorrhage* | *Leukolymphocytic Infiltration* | *Hemorrhage* |
| Control | – | – | – | – |
| Venom | – | ++ | + | + |
| AnV + V (24h) | + | +++ | ++ | – |
| AnV + V (3 days) | ++ | ++ | | |
| AnV + V (7 days) | +++ | + | | |
| *Local Injection of AV* | | | ++ | + |

In the course of recovery 24h after antivenom treatment, it was noticed that the histological structure is obliterated and the tissue is edematous. The lymphoid follicles have undergone mild hyperplasia (+), and the red pulp is markedly congested. The perifollicular areas contain large fields of hemorrhage (severe hemorrhages +++). The stromal vessels are congested, and the capsule is represented by a thin connective tissue (Fig 3E and 3F and Table 2).

At the recovery stage, 3 days post-antivenom treatment, it was observed that the structure of the spleen is obliterated. The tissue is edematous, and the lymphoid follicles have undergone moderate hyperplasia (++). The red pulp is moderately congested. The perifollicular areas contain foci of hemorrhage (moderate hemorrhages ++). The stromal vessels are congested. The capsule is represented by a thin connective tissue (Fig 3G and 3H and Table 2).

In the course of recovery by antivenom treatment (after 7 days), it was noticed that the histological structure was obliterated. The lymphoid follicles have undergone strong hyperplasia (severe hyperplasia +++). The red pulp is moderately congested, and the perifollicular areas contain small foci of hemorrhage (mild hemorrhages +). The stromal vessels are congested. The capsule is represented by a thin connective tissue (Fig 3I and 3J and Table 2).

### 3.4. Experimental antivenom reduce the formation of venom-induced dermal lesions *in vivo*, but not completely

The following experimental animal model was used to assess the preclinical efficacy of the ovine-derived experimental antivenom at preventing the formation of venom-induced dermal lesions. The appropriate intradermal (ID) doses of *M. l. obtusa* venom that elicit the formation of noticeable large dermal lesions without causing any evident systemic intoxication effects were used. Next, the venom was co-incubated with antivenom (or PBS vehicle for control) for 30 minutes at 37 °C, before ID-injecting the venom-plus-antivenom mixture into separate groups of six rats. The exact number of animals were used for the ID administration of the venom, following the avtivenom injection in the same site with 10-minute delay. To allow sufficient time for dermonecrosis to develop fully, the rats were euthanised after 24 hours, after which their skin around the hemorrhagic spot was excised, photographed, and measured. Representative images and the measured results of this experiment are shown in Fig 4. No lesions were observed in the controls and antivenom+venom preincu-bated mixture group (Fig 4B). *M. l. obtusa* venom caused a significant lesion area, which was not completely neutralized by antivenom locally administered in the hemorrhage area 10 minutes after venom injection. However, the hemorrhagic spot was significantly reduced by the experimental AnV, providing evidence in favor of using the antivenom in small doses for local injection in parallel with systemic IV administration to achieve overall neutralization of envenomation symptoms.

### 3.5. *M.l. obtusa* envenomation causes certain histopathological changes in skin tissue even after recovery by antivenom treatment

The skin tissue sections from the control group rats show a normal histological appearance. The histological structure of the skin tissue is preserved. The integrity of the squamous epithelium is maintained, and the keratin layer is developed. There is no inflammatory cell infiltration in the subepithelial tissues (Fig 5A and 5B).

In the venom group, the integrity of the squamous epithelium is preserved, and the keratin layer is developed. The subepithelial tissues are edematous and mildly infiltrated with neutrophils (mild inflammatory reaction +), the blood vessels are hyperemic, and small foci of hemorrhage are present (mild hemorrhages +) (Fig 5C and 5D and Table 2).

In the course of recovery 24h after antivenom treatment (antivenom-venom preincubated), it was noticed that the integrity of the squamous epithelium is preserved, and the keratin layer is sufficiently developed. The subepithelial tissues are edematous and moderately infiltrated with neutrophils and lymphocytes (moderate inflammatory reaction ++). No foci of hemorrhage are present (Fig 5E and 5F and Table 2).

In the course of recovery after local injection of antivenom (with 10-minute delay) treatment after 24h, the integrity of the squamous epithelium is preserved, and the keratin layer is developed. The subepithelial tissues are edematous and show moderate infiltration with neutrophils and lymphocytes (moderate inflammatory reaction ++), and small foci of hemorrhage are present (mild hemorrhages +) (Fig 5G and 5H and Table 2).

## 4. Discussion

Coagulopathy induced by Viperidae venoms involves a complex pathophysiology where venom components disrupt normal hemostasis through multiple mechanisms. These venoms typically contain snake venom metalloproteinases (SVMPs), serine proteases (SVSPs), and other toxins that target clotting factors, endothelial cells, and platelets, leading to consumption coagulopathy, hemorrhage, and vascular damage [34]. The venom can cause both defibrinogenation and activation of coagulation pathways, resulting in unpredictable bleeding and clotting disorders [10,17]. The observation that Prothrombin Time was more prolonged in the antivenom-treated group compared to the venom-only group high-lights the complex interaction between venom action, antivenom intervention, and the coagulation system. A plausible explanation for this phenomenon is that in the absence of antivenom, animals may succumb more rapidly to the systemic effects of venom, including hemorrhage, shock, or organ failure, thereby limiting the observable duration of coagulopathy. Conversely, antivenom administration prolongs survival, allowing the venom-induced damage to coagulation pathways

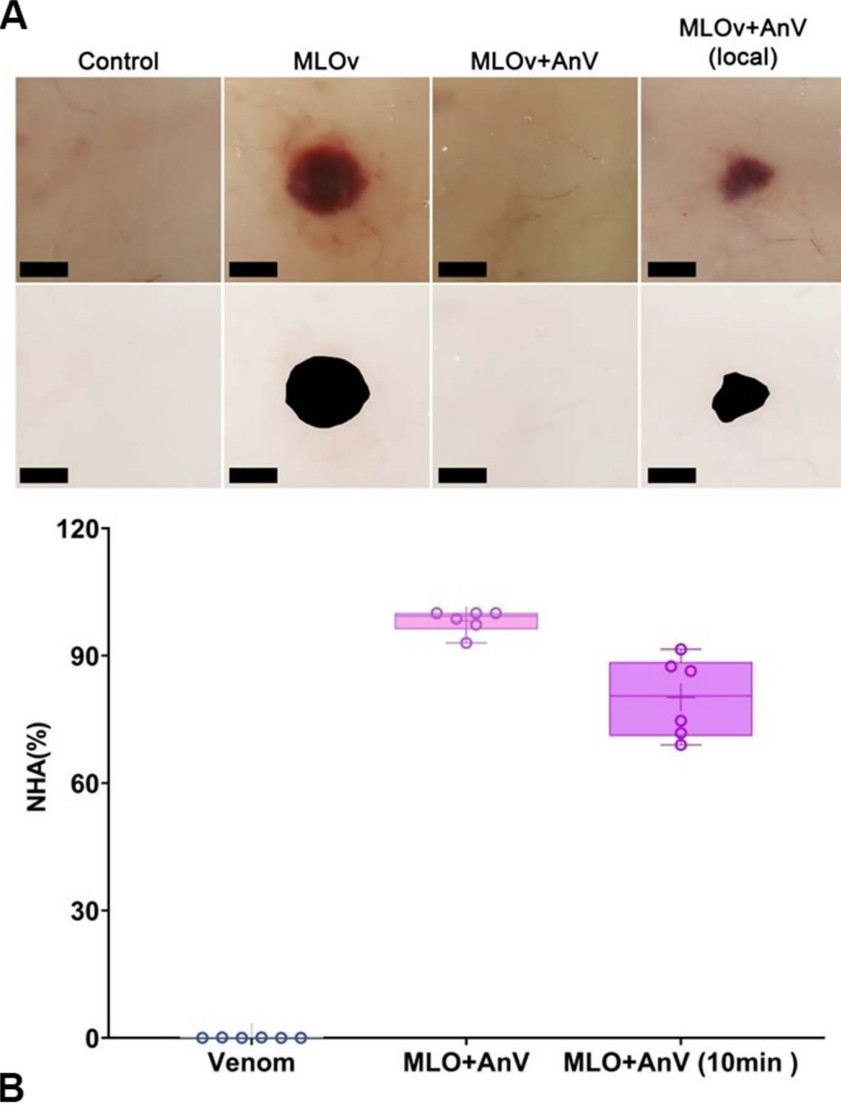

**Fig 4. Dermal lesions induced by *M. l. obtusa* (MLO) venom are inhibited by ovine-derived experimental antivenom. A.** Representative images of the lesions resulting from each treatment group (upper row, black scale bars represent 10mm) and their processing with Photoshop to measure the hemorrhagic spots' size (lower row). **B.** Bar graphs summarizing the average total lesion areas for each treatment group when pre-incubated with venom vehicle control, venom antivenom pre-incubated mixture, and venom following antivenom injection with 10-minute delay (experimental repeats–6, mean±SEM; one-way ANOVA with Dunnett's multiple comparison test).

to continue evolving over a more extended period [7,35]. Thus, while the antivenom neutralizes circulating toxins and prevents immediate death, it does not reverse the structural and functional damage already inflicted upon the coagulation system. Furthermore, the venom may cause irreversible consumption or degradation of clotting factors, endothelial injury, or hepatic dysfunction, all of which contribute to sustained coagulopathy [10,16]. Antivenom therapy cannot immediately replenish depleted clotting factors or repair tissue injury, resulting in a persistent prolongation of Prothrombin Time even after venom neutralization. It is also possible that the formation of venom-antivenom immune complexes or the inflammatory response to venom components could further dysregulate coagulation mechanisms in treated animals [8].

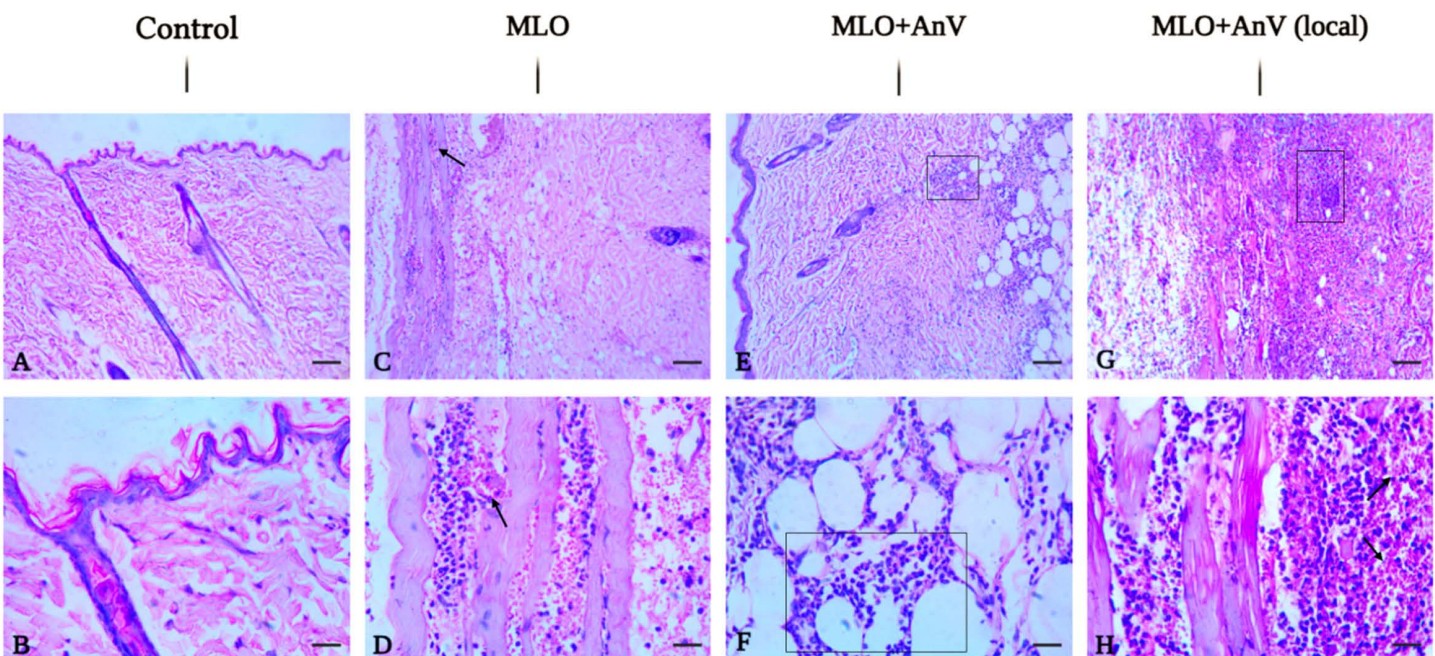

**Fig 5. Histopathological changes in the skin after administration of *M. l. obtusa* (MLO) venom, a venom antivenom mixture, and antivenom injected locally following venom exposure.** The representative field images were acquired with Olympus BX43. Upper row: magnification ×100, scale bar = 250 µm. Lower row: magnification ×400, scale bar = 100 µm. Arrow indicates areas of hemorrhage; Square highlights leukolymphocytic infiltration.

In this context, the greater prolongation of Prothrombin Time observed in the antivenom group is not necessarily indicative of treatment failure but rather reflects the prolonged course of venom-induced coagulopathy that becomes apparent when early death is prevented. These findings emphasize the need for adjunctive therapeutic strategies aimed at supporting the coagulation system during recovery, such as administration of clotting factor concentrates or plasma infusions, in addition to the use of antivenom alone [1].

Overall, these results reveal that while antivenom is crucial for improving survival following envenomation by *M. l. obtusa*, its ability to restore normal coagulation is incomplete, and the full spectrum of venom-induced hemostatic abnormalities may become more evident in the presence of antivenom therapy. Further investigation into the temporal progression of coagulation recovery post-antivenom administration and the potential benefits of supportive hemostatic therapies is warranted to optimize clinical outcomes in cases of severe viper envenomation.

This study also provides important evidence that *M. l. obtusa* envenomation causes persistent and evolving histopathological damage to spleen tissue, even after the administration of antivenom. Despite the widely accepted role of antivenom therapy as the most effective treatment for snakebite envenoming, our findings underscore its limitations in completely reversing the tissue-level consequences of venom toxicity, particularly in lymphoid organs such as the spleen.

The spleen plays a crucial role in immune surveillance and hematopoietic regulation. Under normal conditions, as observed in the control group, splenic architecture is well-preserved with intact connective tissue capsule, well-developed lymphoid follicles, moderately congested red pulp, and no signs of hemorrhage. However, exposure to *M. l. obtusa* venom initiates a cascade of events that progressively disrupt this structure. *M. l. obtusa* venom, like that of other viperid snakes, contains high concentrations of SVMPs and SVSPs, which are known to cause extensive hemotoxicity by damaging vascular integrity, degrading extracellular matrix proteins, and interfering with blood coagulation pathways [13,36]. These effects are particularly evident in the spleen due to its rich vascularization and immune activity.

Following envenomation, spleen tissue becomes edematous and congested, with the first histological signs of hemorrhage appearing in the perifollicular regions. Although the general architecture is only partially compromised at this stage, the damage is already indicative of SVMP-mediated disruption of capillary structures. This corresponds with the known mechanism whereby SVMPs degrade basement membrane proteins and endothelial attachments, leading to capillary rupture and hemorrhage. The red pulp congestion reflects both increased blood flow to the damaged tissue and impaired drainage due to vascular injury [37]. Furthermore, the congestion of stromal vessels likely indicates ongoing vascular leakage and systemic inflammation, typical features of acute hemotoxic envenoming.

Interestingly, the administration of antivenom does not lead to immediate histological improvement. On the contrary, 24 hours post-envenomation, tissue integrity continues to deteriorate, with obliteration of splenic architecture, increased edema, and the appearance of severe hemorrhage in perifollicular areas. This suggests that the antivenom, while neutralizing circulating venom components, may not effectively reach or reverse damage already initiated at the tissue level. At this stage, lymphoid follicles undergo mild hyperplasia, potentially as a compensatory immune response [38]. The severity of hemorrhage and congestion observed post-treatment also shows that venom-induced endothelial and coagulation disturbances persist even after systemic venom neutralization. By day three after antivenom treatment, the spleen remains structurally compromised, with moderate hyperplasia of lymphoid follicles and ongoing hemorrhagic lesions. This indicates a transitional phase where immune activation is escalating, but vascular and stromal repair remain incomplete [39]. Although the intensity of hemorrhage is reduced compared to the 24-hour mark, the persistence of edema and vascular congestion highlights the prolonged nature of venom-induced inflammatory and vascular responses. By day seven, further changes are evident. Hyperplasia of lymphoid follicles becomes severe, reflecting heightened immune activity, likely driven by ongoing inflammation and tissue remodeling. While hemorrhage in the perifollicular areas becomes milder, it remains present, suggesting a slow resolution of vascular injury. These results collectively suggest that while antivenom halts the progression of systemic envenomation, it does not promptly reverse local tissue damage (see Table 2).

These findings resonate with global reports on the challenges of managing snakebite envenomation, particularly those involving viperid species and concerning the drastic irreversible local damages caused by the viper's venom in dermal and muscle tissues [40,41]. This problem is a well-known complication of snakebite cases worldwide and is the most actual health burden in the area of snakebite therapy [42]. Following the old Soviet clinical protocols for the antivenom administration during the *Macrovipera* species-caused envenomations, we tried to use the "challenge then treat" mode of envenomation with a following local injection of antivenom directly in the dermal lesion site. Although in this model, compared to the "classical" assay of the hemorrhagic capacity neutralization, which is based on the WHO-advised preincubation model of the venom-antivenom injection [28,43], the spot of the tissue lesion appears, however, it is much smaller than in the case of an untreated venom lesion area. Moreover, our histological data 24 hours post-treatment also shows less developed local dermal damage and very mild hemorrhage. This can make protocols for the patients with viper venom bites, which will include the local injection of the antivenom dose in parallel with the intravenous administration of the same antivenom for the neutralization of the severe systemic pathological damages of the inner organs and tissues, quite actual.

Therefore, although WHO emphasizes that most snakebite-related deaths and severe complications are preventable with antivenom [3,4,28], our data demonstrate that recovery at the organ level can be protracted and incomplete. In summary, the histopathological evidence from this study highlights the complex pathogenesis of *M. l. obtusa* envenomation and the partial efficacy of antivenom in reversing organ-specific damage. The venom's hemotoxic components initiate structural disintegration that continues even after systemic neutralization, and recovery is marked by persistent edema, hemorrhage, and immune hyperactivation. These results suggest that adjunctive therapies targeting inflammation, endothelial repair, and matrix regeneration may be necessary to achieve complete tissue recovery following envenomation. Our data emphasize the importance of timely intervention and the need for more comprehensive treatment strategies that go beyond venom neutralization to ensure full functional recovery of affected tissues.

Ultimately, our data provides the first empirical evidence that the local damage treatment of the 'rescue' model of envenoming can lead to a new approach for the further experimental investigations of the viper venoms' effect on the dermal and muscle tissues, and thus advocates for the future translation of such combinations of local plus systemic treatments to reduce the life-changing consequences of the viper's snakebite.

## Supporting Information

**S1 Table. Complete blood count (CBC) raw data organized by group and animal.** Groups present in the sheet are: Control, Venom and AnV + V (24hours).
(XLSX)

**S2 Table. Coagulation assay raw data by group and animal.** Groups present in the sheet are: Control, Venom, AnV + V (24hours).
(XLSX)

## Author contributions

**Conceptualization:** Gevorg Avagyan, Heghine Khachatryan, Naira Ayvazyan.

**Data curation:** Heghine Khachatryan, Anna Karapetyan.

**Formal analysis:** Gevorg Avagyan, Vardan Dabaghyan, Ashot Aslanyan, Arsen Kishmiryan.

**Investigation:** Gevorg Avagyan, Heghine Khachatryan, Anna Karapetyan, Naira Ayvazyan.

**Methodology:** Gevorg Avagyan, Vardan Dabaghyan, Ashot Aslanyan, Arsen Kishmiryan.

**Supervision:** Naira Ayvazyan.

**Validation:** Naira Ayvazyan.

**Visualization:** Gevorg Avagyan, Naira Ayvazyan.

**Writing – original draft:** Gevorg Avagyan.

**Writing – review & editing:** Naira Ayvazyan.

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
