## [Decision Letter · Decision Letter 0]

15 Oct 2025

Hematological and coagulation alterations and splenic response following Macrovipera lebetina obtusa envenomation: Evaluation of ovine-derived experimental antivenom

Dear Dr. Avagyan,

Thank you for submitting your manuscript to PLOS Neglected Tropical Diseases. After careful consideration, we feel that it has merit but does not fully meet PLOS Neglected Tropical Diseases's publication criteria as it currently stands. Therefore, we invite you to submit a revised version of the manuscript that addresses the points raised during the review process.

Please submit your revised manuscript within 60 days Dec 14 2025 11:59PM. If you will need more time than this to complete your revisions, please reply to this message or contact the journal office at plosntds@plos.org. Please include the following items when submitting your revised manuscript:

We look forward to receiving your revised manuscript.

Kind regards,

Wuelton Monteiro, Ph.D.

Section Editor

Wuelton Monteiro

Section Editor

Shaden Kamhawi

co-Editor-in-Chief

Paul Brindley

co-Editor-in-Chief

**Journal Requirements:**

1) Please upload all main figures as separate Figure files in .tif or .eps format. For more information about how to convert and format your figure files please see our guidelines: 

2) We have noticed that you have uploaded Supporting Information files, but you have not included a list of legends. Please add a full list of legends for your Supporting Information files after the references list.

3) Please ensure that the funders and grant numbers match between the Financial Disclosure field and the Funding Information tab in your submission form. Note that the funders must be provided in the same order in both places as well.

**Reviewers' Comments:**

Reviewer's Responses to Questions

**Key Review Criteria Required for Acceptance?**

**Methods:**

-Are the objectives of the study clearly articulated with a clear testable hypothesis stated?

-Is the study design appropriate to address the stated objectives?

-Is the population clearly described and appropriate for the hypothesis being tested?

-Is the sample size sufficient to ensure adequate power to address the hypothesis being tested?

-Were correct statistical analysis used to support conclusions?

-Are there concerns about ethical or regulatory requirements being met?

Reviewer #1: 1- Could the ambient temperature during vacuum drying of the venom have affected the enzymatic activity?

2- Considering that antivenom administration can mitigate some systemic effects, but not the local effect, what is the rationale for intradermal injection of antivenom 10 minutes after venom injection?

Reviewer #2: 1. Hypothesis Statement

While the objectives are clearly articulated and logically derived from the contextual background, the manuscript does not explicitly present a formal, testable hypothesis within the Introduction. Articulating such a hypothesis would strengthen the scientific rigor and focus of the study.

2. Study Design

The study design is comprehensive and methodologically robust, aligning well with the stated objectives of characterizing the effects of Macrovipera lebetina obtusa venom and assessing antivenom efficacy. The chosen methodologies are appropriate for addressing the research questions posed.

3. Study Population

The study population is clearly described and well-suited to the research aims. The selection criteria and characteristics of the population are appropriate for investigating the effects of envenomation and antivenom intervention in the context of neglected tropical diseases.

4. Sample Size and Statistical Power

However, the reported sample size appears insufficient to ensure adequate statistical power for robust hypothesis testing, particularly given the variability inherent in toxinology and neglected tropical disease research. Increasing the sample size would enhance the reliability and generalizability of the findings.

5. Statistical Analysis

The statistical analyses employed are appropriate and correctly applied to the research questions and experimental design. However, the authors did not explicitly address the assumptions underlying the use of ANOVA. A discussion of these assumptions and how they were verified would improve the transparency and validity of the statistical approach.

6. Ethical and Regulatory Compliance

The study appears to adhere to all major ethical and regulatory requirements, with evidence of rigorous oversight and compliance with international standards for research involving potentially hazardous biological materials.

**Results:**

-Does the analysis presented match the analysis plan?

-Are the results clearly and completely presented?

-Are the figures (Tables, Images) of sufficient quality for clarity?

Reviewer #1: Would be valuable if the authors add to Table 1 the measurements obtained from different timepoint (i.e., 1st day, 3rd day, and 7th day).

The authors state that " One clotting measure, the prothrombin time, stayed abnormal even after antivenom treatment, showing that the venom’s effects were only partly neutralized." The neutralization limitation of different antivenom formulations (complete IgG, F(ab’)2, etc.) is a common question in toxinology. I recommend that the authors discuss the secondary effect of SVMP and SVSP on hemostatic parameters. Since SVSP activates the coagulation cascade, antivenom is expected to have no effect on hemostatic parameters directedly dependent on the toxin’s activity.

Reviewer #2: 1. Alignment of Analyses with Methodology

The analyses presented in the manuscript are consistent with the analysis plan outlined in the methodology section. However, a minor limitation is the absence of explicit identification of the statistical tests employed in the results section. Clearly naming these tests would enhance transparency and reproducibility.

2. Addressing Outcomes and Contextualization

All primary outcomes described in the methods are thoroughly addressed in the results, and the findings are effectively contextualized within the broader objectives of the study. This approach strengthens the manuscript’s contribution to the field and situates the results within the current landscape of Neglected Tropical Diseases research.

3. Quality of Tables and Figures

The tables and figures included in the manuscript are of high quality and clarity, meeting the standards expected in leading journals focused on Neglected Tropical Diseases. Their presentation facilitates comprehension and supports the effective communication of key findings.

**Conclusions:**

-Are the conclusions supported by the data presented?

-Are the limitations of analysis clearly described?

-Do the authors discuss how these data can be helpful to advance our understanding of the topic under study?

-Is public health relevance addressed?

Reviewer #1: The authors suggest that local injection of antivenom would have clinical value. In clinical settings, the patient arrives hours after the bite. How could an intervention 10 minutes after venom challenge have translational potential?

Reviewer #2: 1. Strength of Conclusions

The study’s conclusions are clearly and directly supported by the data presented. The evidence for hemoconcentration, coagulopathy, and persistent tissue damage is robust and well-documented. The nuanced interpretation of the partial efficacy of antivenom, along with the call for adjunctive therapies, is well justified by the results and aligns with current scientific understanding in the fields of Neglected Tropical Diseases and viper envenomation.

2. Discussion of Limitations

The authors have appropriately acknowledged the biological and therapeutic limitations of antivenom treatment, as well as the complexity of venom pathophysiology. However, the manuscript would benefit from a more explicit and structured discussion of analytical and methodological limitations, including sample size, statistical assumptions, and the inherent constraints of animal models. Addressing these aspects would further strengthen the transparency and rigor of the study.

3. Advancement of Knowledge

The discussion clearly articulates how the data advance our understanding of viper envenomation and its management. The authors thoughtfully situate their findings within the broader context of Neglected Tropical Diseases, highlighting the implications for both research and clinical practice.

4. Translational Relevance

The manuscript effectively bridges the gap between bench research and its implications for health systems, clinical practice, and policy in the context of Neglected Tropical Diseases. This translational perspective is particularly valuable given the ongoing challenges in addressing these diseases in resource-limited settings.

**Editorial and Data Presentation Modifications?**

Reviewer #1: Minor reviews:

Use Macrovipera lebetina obtusa on the first use and then abbreviate as M. l. obtusa or M. lebetina. Avoid the MLO nomenclature as it is not a good practice.

In line 33, correct the “it unable” to “it was unable”.

In line 98, close the parenthesis in anticoagulation.

In line 115, define CBC at first use. Afterward use the abbreviation instead of repeating.

In line 125, use italic for Macrovipera lebetina obtusa.

In line 151, add a space in the between the number and the unit “(24 hours)” for consistence.

In line 189 use “euthanized” instead of “sacrificed”.

The Figure 4 need the A and B marks in the panels. Additionally, the legend needs a correction In line 356. Check the following sentence: “antivenom injection with 10 10-minute delay.”

Legends of Figures 3 and 5 are confusing. The legends describe the magnification for upper panel as 400x and for the lower panel as 100x, but the image seems to be the opposite.

Place the Table 2 at the results where it is first mentioned.

Reviewer #2: In addition to the comments I have included in this form, I have uploaded an annotated version of the article.

**Summary and General Comments**

Reviewer #1: The work by Avagyan and colleagues demonstrates how Macrovipera lebetina obtuse venom affects hemostatic parameters in a murine model. The study demonstrates the effects of venom injection on coagulation parameters and spleen histology. The work is valuable, and this reviewer has some suggestions for improving the manuscript. First, the lack of substantial references supporting the discussion of the results is notable. In fact, there are almost no references in the discussion. The main results presented in the study are expected and well described for several other viperid snakes. Authors should include appropriate literature in their discussion.

Reviewer #2: This article represents a significant and well-executed contribution to the fields of Neglected Tropical Diseases and toxinology. Its strengths include a comprehensive experimental approach, methodological rigor, and clear, transparent reporting. The study’s focus on a regionally important viper species and the evaluation of a novel antivenom yield new and actionable insights for both research and clinical practice. The article exemplifies high standards of scholarship and is poised to inform future research, policy, and clinical management of snakebite envenomation in regions endemic for neglected tropical diseases.

PLOS authors have the option to publish the peer review history of their article (what does this mean? ). If published, this will include your full peer review and any attached files.

**Do you want your identity to be public for this peer review?** For information about this choice, including consent withdrawal, please see our Privacy Policy .

Reviewer #1: **Yes: ** Alison FA Chaves

Reviewer #2: **Yes: ** Antonio Marcus Nogueira Lima

**Figure resubmission:**
---

## [Editor Report · Decision Letter 1]

4 Nov 2025

Dear Mr. Avagyan,

We are pleased to inform you that your manuscript 'Hematological and coagulation alterations and splenic response following Macrovipera lebetina obtusa envenomation: Evaluation of ovine-derived experimental antivenom' has been provisionally accepted for publication in PLOS Neglected Tropical Diseases.

Best regards,

Wuelton Monteiro, Ph.D.

Section Editor

Wuelton Monteiro

Section Editor

Shaden Kamhawi

co-Editor-in-Chief

Paul Brindley

co-Editor-in-Chief

---

## [Editor Report · Acceptance letter]

Dear Mr. Avagyan,

We are delighted to inform you that your manuscript, " 

Hematological and coagulation alterations and splenic response following Macrovipera lebetina obtusa envenomation: Evaluation of ovine-derived experimental antivenom," has been formally accepted for publication in PLOS Neglected Tropical Diseases.

Best regards,

Shaden Kamhawi

co-Editor-in-Chief

Paul Brindley

co-Editor-in-Chief
